# Stress Watch: The Use of Heart Rate and Heart Rate Variability to Detect Stress: A Pilot Study Using Smart Watch Wearables

**DOI:** 10.3390/s22010151

**Published:** 2021-12-27

**Authors:** Taryn Chalmers, Blake Anthony Hickey, Phillip Newton, Chin-Teng Lin, David Sibbritt, Craig S. McLachlan, Roderick Clifton-Bligh, John Morley, Sara Lal

**Affiliations:** 1Neuroscience Research Unit, School of Life Sciences, University of Technology Sydney, Broadway, Sydney, NSW 2007, Australia; Taryn.Chalmers@uts.edu.au (T.C.); blake.hickey1@my.nd.edu.au (B.A.H.); 2School of Nursing and Midwifery, Western Sydney University, Penrith, NSW 2747, Australia; P.Newton@westernsydney.edu.au; 3Australian AI Institute, University of Technology Sydney, Broadway, Sydney, NSW 2007, Australia; Chin-Teng.Lin@uts.edu.au; 4School of Public Health, University of Technology Sydney, Broadway, Sydney, NSW 2007, Australia; David.Sibbritt@uts.edu.au; 5Centre for Healthy Futures, Torrens University, Sydney, NSW 2009, Australia; craig.mclachlan@laureate.edu.au; 6School of Medicine, University of Sydney, Sydney, NSW 2006, Australia; roderick.cliftonbligh@sydney.edu.au; 7School of Medicine, Western Sydney University, Penrith, NSW 2747, Australia; J.Morley@westernsydney.edu.au

**Keywords:** FitBit, stress, smart technology, heart rate variability, anxiety, depression, wearable device

## Abstract

Stress is an inherent part of the normal human experience. Although, for the most part, this stress response is advantageous, chronic, heightened, or inappropriate stress responses can have deleterious effects on the human body. It has been suggested that individuals who experience repeated or prolonged stress exhibit blunted biological stress responses when compared to the general population. Thus, when assessing whether a ubiquitous stress response exists, it is important to stratify based on resting levels in the absence of stress. Research has shown that stress that causes symptomatic responses requires early intervention in order to mitigate possible associated mental health decline and personal risks. Given this, real-time monitoring of stress may provide immediate biofeedback to the individual and allow for early self-intervention. This study aimed to determine if the change in heart rate variability could predict, in two different cohorts, the quality of response to acute stress when exposed to an acute stressor and, in turn, contribute to the development of a physiological algorithm for stress which could be utilized in future smartwatch technologies. This study also aimed to assess whether baseline stress levels may affect the changes seen in heart rate variability at baseline and following stress tasks. A total of 30 student doctor participants and 30 participants from the general population were recruited for the study. The Trier Stress Test was utilized to induce stress, with resting and stress phase ECGs recorded, as well as inter-second heart rate (recorded using a FitBit). Although the present study failed to identify ubiquitous patterns of HRV and HR changes during stress, it did identify novel changes in these parameters between resting and stress states. This study has shown that the utilization of HRV as a measure of stress should be calculated with consideration of resting (baseline) anxiety and stress states in order to ensure an accurate measure of the effects of additive acute stress.

## 1. Introduction

Stress is an inherent part of the normal human experience. In its most primal sense, the provocation of a stress response provides a valuable mechanism by which an organism can respond appropriately to internal and external stressors. Although, for the most part, this stress response is advantageous, chronic, heightened, or inappropriate stress responses can have deleterious effects on the human body. Chronic stress has been linked to numerous disease states, from cardiovascular disease and stroke [1] to irritable bowel disease [2] and cancer [3,4,5,6]. Research has also shown acute stress can have deleterious effects on brain functioning [7,8,9], cardiovascular health [10,11], and gut functioning [12,13]. Further, there remains a significant link between biologically inappropriate stress responses and mental health, with individuals who experience chronic stress significantly more likely to suffer from anxiety [14,15] and depression [16,17].

Certain populations find themselves at higher risk of stress. For instance, medical students have often been shown to experience significantly more stressful events than the general population [18]. The potential negative effects of emotional distress on medical students include reductions in educational outcomes and impairment in clinical capability when stress-induced states compromise decision making capabilities [19]. However, studies have also shown that individuals in fields such as medicine are often more adept at self-regulating stress responses, and thus exhibit a reduced stress reactivity when compared to the general population [20,21]. That is, populations which find themselves exposed to regular stressful events often report lower levels of perceived baseline stress and a diminished stress response to external stressors [22]. Further, the stressful events experienced by clinical year medical students has been shown to be comparable to that of other professions, such as defense force personnel, police officers, and paramedics [23]. It has been suggested that individuals who experience repeated or prolonged stress (such as medical students, police officers, etc.) exhibit blunted biological stress responses when compared to the general population [22]. Thus, when assessing whether a ubiquitous stress response exists, it is important to stratify based on resting levels in the absence of stress. Research has shown that stress that causes symptomatic responses requires early intervention in order to mitigate possible associated mental health decline and personal risks [24]. Given this, real-time monitoring of stress may provide immediate biofeedback to the individual and allow for early self-intervention.

The concept of allostasis is of particular importance, with allostasis reflecting an individual’s physiological response to stressful situations and maintenance of homeostasis by how an individual adapts to stressors. It is of importance as it supports the aforementioned theory that immersion in frequent stressful scenarios associated with certain careers may result in an individual becoming more capable of responding to stress; however, allostatic load (the concept of ongoing failed adaptation to stress) can present in individuals who do not respond well to stress/have poor resilience and are at higher risk of the chronic illnesses mentioned above due to poorer physiological responses to stress [25,26]. This study aimed to determine if the change in heart rate variability could predict, in two different cohorts, the quality of response to acute stress when exposed to an acute stressor and, in turn, contribute to the development of a physiological algorithm for stress which could be utilized in future smartwatch technologies. This study also aimed to assess whether baseline stress levels may affect the changes seen in heart rate variability at baseline and following stress tasks. It is hypothesized that low frequency HRV will increase and HF HRV will reduce during the stress task. It is also hypothesized that the medical population will exhibit reduced stress induced HRV excitability when compared with the general population.

## 2. Materials and Methods

### 2.1. Participants and Sampling

The current study utilized a cross-sectional study design. Participants in the current analysis included 30 medical student participants in their clinical year and 30 normative participants aged between 24 and 58 years. This study was undertaken in a light- and sound-controlled laboratory environment. Prior to inclusion, an in-house designed lifestyle questionnaire was used to screen participants for medication use (participants who regularly consumed medications which alter HRV were excluded), alcohol intake, smoking habits, and chronic disease/illness. Participants were excluded if they smoked more than 10 cigarettes per day or had smoked in the previous 12 h, had consumed alcohol in the previous 12 h, or were taking any prescription medication known to affect heart rate or blood pressure. Participants provided informed consent prior to beginning the study. Two participants were excluded, one due to taking a calcium-channel blocker and one due to the inability to remain for the duration of the study.

### 2.2. Experimental Procedure

At the commencement and conclusion of each session, the participant’s blood pressure (BP) was measured three times with an automated blood pressure monitor (Omron IA1B, Japan) and then averaged to derive an average BP value. Following BP measurement, a purpose designed questionnaire was used to collect demographic, lifestyle, and work-related data. Additionally, the general health questionnaire (GHQ) was utilized to assess general psychological wellbeing whilst the 42-item Depression Anxiety Stress Scale (DASS) was applied to assess symptom severity of stress. At present, interpretation of the DASS is based primarily on the use of cut-off scores. Severity is stratified by ratings from ‘normal’ to ‘extremely severe’ on the basis of percentile scores, with 0–78 classified as ‘normal’, 78–87 as ‘mild’, 87–95 as ‘moderate’, 95–98 as ‘severe’, and 98–100 as ‘extremely severe’ [27].

Following completion of the questionnaires, participants had a 3-lead ECG fitted. The participant was also fitted with a FitBit Versa 2 on their non-dominant hand. The FitBit was used to record second-by-second heart rate data.

The Trier Social Stress Test (TSST) [28] was then utilized to elicit a controlled stress response in participants. For this section of the study, participants underwent a 15 min resting baseline session followed by the TSST, which constituted of a 5 min preparation/anticipation task where participants were required to prepare a short speech, followed by a 5 min public speaking task, and, finally, a 5 min mental arithmetic task. The Trier Social Stress Test (TSST) is one of the most well-accepted laboratory techniques to induce acute stress in experimental settings and has been repeatedly shown to reliably provoke hypothalamic–pituitary–adrenal axis activation [28,29,30,31,32,33,34] and is particularly useful for studies of stress assessment and stress hormone reactivity [30].

During baseline and each component of the TSST, 3-lead ECG data were captured using disposable Ag/AgCl electrodes placed on the participant’s upper torso under each clavicle on the coracoid processes and one just below the sternum over the xiphoid process [35]. Following completion of the TSST, three further BP reading were obtained which were then averaged.

### 2.3. ECG Data Processing

Raw ECG data were processed utilizing the Kubios HRV Premium software (Version 3.1.0, Kubios Oy, Kuopio, Finland) to generate HRV parameters. Frequency domain HRV deconstructs the variance in beat-to-beat HR into its underlying components at different frequencies using fast-Fourier transforms (FFTs). Note that the calculation of frequency domain HRV requires a certain level of ‘stationarity’ (i.e., the mean and variance of the ECG signal do not vary significantly at different points during the electrocardiogram). In order to meet this requirement, data were collected over 5-min intervals and averaged to produce reliable HRV frequency domain data. Further, all data were assessed to ensure sinus rhythm prior to transformation. Frequency domain activity was calculated using Welch’s periodogram method [36] for the following HRV frequency bands: low frequency (LF) power (0.04–0.15 Hz), high frequency (HF) power (0.15–0.4 Hz), and total power (TP), as well as the LF:HF ratio. Note that, in the process of deriving these variables, the automatic artefact correction process of the Kubios software [37] as well as the Smoothn Priors method of trend component rejection were utilized, and HRV data were log-transformed prior to analysis, where relevant.

### 2.4. FitBit Data Processing

FitBit was used to record heart rate data alone. Heart rate variability data are not yet commercially available from the FitBit. ECG derived HRV data were correlated with the FitBit data to work toward an algorithm.

### 2.5. Statistical Analysis

Statistical analysis was performed using SPSS Version 23.0 (SPSS, Inc.; Chicago, IL, USA) with statistical significance defined as *p* < 0.05. Data were initially subject to descriptive statistics. A paired t-test was used to determine significant differences in HRV parameters between the resting phase and stress task. As multiple comparisons were made, a Holm–Bonferroni correction was then applied to avoid type I errors with the family-wise α (a) level set at 0.05. Following that, partial Pearson’s correlation analysis (controlling for age and body mass index (BMI)) was utilized to determine associations between HRV parameters and DASS scores during the baseline and stress task (TSST).

## 3. Results

### 3.1. Participant Descriptives

This study cohort consisted of 60 participants (33 males and 27 females), with demographic information provided in Table 1. The average age of participants was 28.9 ± 8.8 (years ± SD), with an average body mass index of 23.1 ± 3.4 kg/m^2^. The two study groups were age (*p* = 0.36) and weight (*p* = 0.26) matched. Within the medical group, the mean systolic blood pressure pre-study was 117 ± 13 mmHg, which was significantly higher following the stress task at 120 ± 11 mmHg (*p* = 0.047); with mean diastolic blood pressure pre-study of 76 ± 7 mmHg, which was also significantly higher following the stress task at 80 ± 9 mmHg (*p* < 0.001). Within the general population group, the mean systolic blood pressure pre-study was 115 ± 15 mmHg, which was significantly higher following the stress task at 119 ± 15 mmHg (*p* = 0.014); with mean diastolic blood pressure pre-study of 75 ± 10 mmHg, which was also significantly higher following the stress task at 81 ± 8 mmHg (*p* < 0.001).

### 3.2. Psychometric Data

Participants completed the DASS, with scores for each of the populations found in Table 2. For the stress subscale of the DASS, the average score for the medical group was 11.1 ± 8.9 (score ± SD), which is classified as mild stress, and 20.9 ± 14.2 (score ± SD) for the general population, which is classified as moderate stress. For the depression subscale of the DASS, the average score for the medical students was 2.2 ± 2.6 (score ± SD), which is considered normal, and 13.0 ± 16.8 (score ± SD) for the general population, which is classified as moderate depression. For the anxiety subscale of the DASS, the average score for the medical students was 4.5 ± 6.1 (score ± SD), which is considered normal, and 8.6 ± 7.4 (score ± SD) for the general population, which is classified as mild anxiety. The general population has significantly higher scores for all of the DASS subscales than the medical students (stress: *p* = 0.002; depression: *p* = 0.026; anxiety: *p* = 0.003).

### 3.3. Heart Rate Data

Both resting and stress phase HR were significantly higher in the medical student population than the general population (resting *p* < 0.001; stress *p* = 0.022). Further, both populations had a significant rise in average heart rate between the resting and stress stages (general population, *p* = 0.07; medical student cohort, *p* = 0.04) (Table 3).

### 3.4. Heart Rate Variability Data

For the general population, there was a significant rise in LF (*p* = 0.006), HF (*p* < 0.001), and LF:HF ratio (*p* = 0.007) during the stress task. For the medical student cohort, there was a significant rise in VLF (*p* = 0.023), LF (*p* = 0.031), and HF (*p* = 0.029) during the stress task (Table 4).

When controlling for age and BMI, the anxiety subscale of the DASS showed significant correlations. Anxiety (as measured by the anxiety subscale of the DASS) was found to be correlated to baseline VLF (r = 0.43; *p* = 0.003) and, during the stress task, was correlated to LF (r = −0.49; *p* = 0.0113) and LF:HF ratio (r = 0.46; *p* = 0.02) (Table 5).

## 4. Discussion

The present study aimed to determine if HRV could be used to predict the quality of response to acute stress, comparing the general population to medical students, with the intent to contribute data which may aid in developing a physiological algorithm for stress that could be incorporated into wearable technologies. Whilst there are devices in development capable of detecting stress [38], there is currently no rigorously validated algorithm for stress detection. The current study found a significant rise in LF, HF, and the LF:HF ratio, with HF (*p* < 0.001) and LF:HF ratio during the stress task with medical students having a narrower increase in HRV parameters than the general population. HRV parameters were, however, not significantly correlated to psychometric stress testing, with repeat analysis required with larger cohorts.

The present study found that LF increases significantly following an acute stressor in both the general population and medical student cohorts, reflective of increase sympathetic nervous system activity [39]. Further, LF was significantly negatively correlated with anxiety as reported by the objective DASS score, which further supports the relationship between LF and acute stress. The present study found that baseline depression, anxiety, and stress scores were significantly higher in the general population than those reported by the medical student population. This is a rather novel finding in that literature often reports that medical students experience significantly higher levels of stress than the general population [40,41,42,43,44].

Repeated exposure to stressful events leads to a biological tailoring of the stress response. This often results in blunted physiological stress responses and a reduced stress reactivity [45,46,47]. It has been hypothesized that, amongst other academic traits, the development of this reduced stress reactivity throughout a medical student’s studies, and into their medical career, proved advantageous in both clinical and personal settings [48].

Both the general population and the medical student group responded to a stressful task similarly, with increases in heart rate during the stress task when compared to the resting phase. It is well accepted in the literature that increases in heart rate during a stressful task are biologically advantageous [49,50,51]. As the heart rate increases, so too does the cardiac output which, in turn, increases oxygen perfusion to large peripheral muscle groups. Interestingly, the medical group showed higher average heart rates during both the resting and stress task than the general population. Research has suggested anticipation causes a rise in both resting and stress state heart rate as a compensatory mechanism for the perceived imminent stressor [52]. Certain personality traits have been shown to exhibit higher levels of anticipatory cardiac elevation during laboratory induced stress tasks.

The changes in heart rate variability between the resting and stress tasks both had some overlap in HRV parameters common to both groups as well as some differences. The general population showed significantly higher LF, HF, and TP in the stress task, when compared to baseline, while the medical students showed significantly higher VLF, LF, and HF during the stress task compared to baseline. The common finding across both groups, however, was an increase in LF and HF during the stress task. The literature is in disagreement regarding the impacts of acute stress on LF and HF. Some studies report an increase in LF during stress tasks [53,54,55] whilst others report a decrease in LF during clinically induced stress [56,57]. There remains, however, general consensus regarding the effects of acute mental stress on HF HRV. Studies generally support a decrease in HF HRV during acute mental stressors, owing to a decrease in parasympathetic-oriented concentration during stress tasks [39,58]

Given that LF is generally accepted as a correlate of parasympathetic and sympathetic nervous system activity, our findings are in line with current literature, which suggests an increase in total autonomic arousal during stress by the sympathetic nervous system. Further, high levels of baseline anxiety have been shown to positively correlate to stress reactivity, and thus HRV reactivity, during stress tasks [59].

## 5. Conclusions

The use of biological algorithms to assess HR and HRV stress responses is a novel use of currently available technology. The aim of this study was to determine if average heart rate, and heart rate variability, could be used to identify distinct physiological changes during a stressful task. Although the present study failed to identify ubiquitous patterns of HRV and HR changes during stress, it did identify novel changes in these parameters between resting and stress states.

Small cohort size is a limitation of this study, which limits the generalizability of the results, with further research required with a larger cohort of participants with stratified anxiety scores based off the DASS to further evaluate the correlation with HFV. With respect to study environment, there may be an inherent bias benefiting the medical student group who are familiar with the clinical environment the investigation was conducted in; thereby, relative stress may have been lower at baseline than compared with the general population. This study has shown that the utilization of HRV as a measure of stress should be calculated with consideration of resting (baseline) anxiety and stress states in order to ensure an accurate measure of the effects of additive acute stress.

## Figures and Tables

**Table 1 sensors-22-00151-t001:** Demographics and blood pressure data for the cohort and per grouped division (mean ± standard deviation with range in parentheses). Comparisons are between medical student and general population (Student’s unpaired *t*-tests).

Measure	Cohort (*n* = 60)	Medical Students (*n* = 30)	General Population (*n* = 30)	*p*-Value
Male Gender (%)	55.0 (Male 33; Female 27)	53.3 (Male 16; Female 14)	56.7 (Male 17; Female 13)	-
Age (years)	28.9 ± 8.8	27.8 ± 2.9	29.9 ± 10.3	0.36
Height (cm)	174.5 ± 9.8	173.5 ± 9.9	175.4 ± 9.7	0.53
Weight (kg)	72.7 ± 14.8	70.7 ± 11.9	75.2 ± 17.4	0.26
BMI (kg/m^2^)	23.1 ± 3.4 (18–35)	23.4 ± 2.7 (19–28)	21.5 ± 7.7 (18–35)	0.36
Pre-Study SBP (mmHg)	116 ± 14 (78–143)	117 ± 14 (92–143)	115 ± 15 (78–138)	0.63
Post-Study SBP (mmHg)	120 ± 14 (76–149)	120 ± 11 (102–149)	119 ± 15 (76–141)	0.81

**Table 2 sensors-22-00151-t002:** Scores obtained for depression, anxiety, and stress scale (mean ± standard deviation). Comparisons are between medical students and general population (Student’s unpaired *t*-tests). Normative values for each subscale are presented (Crawford and Henry 2003).

Cohort	Stress Score	Depression Score	Anxiety Score
	(Average ± SD)	(Average ± SD)	(Average ± SD)
Medical Students	11.1 ± 8.9	2.2 ± 2.6	4.5 ± 6.1
General Population	20.9 ± 14.2	13.0 ± 16.8	8.6 ± 7.4
*p*-value	0.002	0.026	0.003
Normative Values	Normal 0–10	Normal 0–9	Normal 0–6
	Mild: 11–18	Mild: 10–12	Mild: 7–9
	Moderate: 19–26	Moderate: 13–20	Moderate: 10–14
	Severe: 27–34	Severe: 21–27	Severe: 15–19

**Table 3 sensors-22-00151-t003:** Average heart rate for the baseline and stress phases, general population vs. medical cohort, which was recorded with FitBit Versa 2.

	General Population	Medical Students	*p*-Value
Resting HR	75.29	79.94	0.07
Stress HR	83.71	90.55	0.04 *
*p*-value	<0.01 *	0.022 *	-

* = significant (*p* < 0.05).

**Table 4 sensors-22-00151-t004:** Comparison between heart rate variability parameters collected from 3-lead ECG during the baseline (resting) and stress tasks for the general population and medical cohorts (mean values).

	HRV Parameter	Baseline	Stress	*p*-Value
General Population	VLF	4.56	4.54	0.916
	LF	53.12	54.62	0.006 *
	HF	46.88	56.29	<0.001 *
	TP	17.22	17.24	0.980
	Ratio	1.14	1.21	0.007 *
Medical Students	VLF	4.59	5.69	0.023 *
	LF	54.48	59.70	0.031 *
	HF	45.52	49.14	0.029 *
	TP	17.26	20.70	0.197
	Ratio	1.21	1.22	0.554

* = significant (*p* < 0.05).

**Table 5 sensors-22-00151-t005:** Correlation between anxiety subscale (of the DASS tool) and heart rate and heart rate variability data in the general population; with heart rate data recorded with FitBit Versa 2 and heart rate variability data collected from 3-lead ECG.

		Correlation Coefficient (*r*)	*p*-Value
Anxiety subscale	Baseline VLF	0.43	0.033 *
	Baseline LF	0.35	0.084
	Baseline HF	−0.35	0.084
	Baseline TP	0.35	0.086
	Baseline Ratio	−0.35	0.090
	Stress LF	−0.49	0.013 *
	Stress Ratio	0.46	0.020 *

* = significant (*p* < 0.05).

## Data Availability

The data presented in this study are available on request from the corresponding author. The data is not publicly available currently due to storage on site at the University of Technology, as stipulated in the Ethics application.

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
