# Peer review of "Stress Watch: The Use of Heart Rate and Heart Rate Variability to Detect Stress: A Pilot Study Using Smart Watch Wearables"

_sensors, 2021, doi:10.3390/s22010151_

Round 1

Reviewer 1 Report

I was very interested in the article based on the title "Stress Watch: The use of heart rate and heart rate variability to detect stress; a pilot study using FitBit Technology." Based on the title, I expected an analysis comparing the use of FitBit-derived data to measures from clinical instruments (i.e. the three-lead ECG). However, this was not the case.  The manuscript makes minor reference to the use of the FitBit to collect heart rate data. The main conclusion from the paper is that any algorithms developed for the use in wearable technologies to provide biofeedback on stress responses should incorporate a comparison of measures across time so allow for assessment of within-wearer changes.

The manuscript could be improved by attending to the following:

Introduction.

The manuscript could be strengthened by including background on the topic of allostasis and allostatic load, topics which support the authors' emphasis on individual differences in stress responses.

Materials and Methods

The authors use various terms to refer to the medical student cohort (e.g. student doctor participants, clinical cohort, medical cohort) which is confusing. Please, select one term to describe the cohort in the text and figures.

Line 102. Check for grammatical errors (e.g. "questionnaires" instead of "questionnaire"

Line 103. "also" is redundant given the sentence starts with "Additionally".

Lines 110-113 seem to be redundant with lines 123-127.

Line 140. Although the LH:HF ratio is a commonly reported metric, the value of LH:HF ratio is questionable and very dependent on context.  See. Shaffer, F. and J. P. Ginsberg (2017). "An Overview of Heart Rate Variability Metrics and Norms." Front Public Health 5: 258. https://www.ncbi.nlm.nih.gov/pubmed/29034226

There is no description about how the FitBit data were downloaded and processed.

There is no explanation of why the heart rate data was collected from the FitBits when the same data could be collected from the ECG data.

There is no comparison between the HR data collected by the FitBit and the ECG, which would help validate the use of FitBits relative to a clinically validated measurement method (the ECGs).

Discussion

The discussion should begin by focusing on the main purpose of the article, which was to inform the development of an algorithm.  The data comparing the medical student and general population cohorts are of interest, but are not the primary purpose of the article.

Lines 246-248. The comment "studies generally support an increase in HF HRV during acute mental stressors" requires a citation, especially since other meta-analyses indicate that there is a decrease in HF HRV indicating a decrease in PNS activity during acute stress. See Kim, H.-G., E.-J. Cheon, D.-S. Bai, Y. H. Lee and B.-H. Koo (2018). "Stress and Heart Rate Variability: A Meta-Analysis and Review of the Literature." Psychiatry Investig 15(3): 235-245. https://pubmed.ncbi.nlm.nih.gov/29486547
https://www.ncbi.nlm.nih.gov/pmc/articles/PMC5900369/

Author Response

Please see attached.
Thank you kindly to the Reviewer for your critiques and expert opinion. It always highly valuable to learn from Academics in the field and your time is greatly appreciated.

Reviewer 2 Report

This is a study investigating a significant subject of acute and chronic stress.

I have a suggestion about the aim which is "to determine whether heart rate variability and average heart rate could be used as a predictor of acute stress". I think it is more like HRV could predict the quality of response to acute stress not acute stress itself.

I would also suggest to elaborate more on the main finding which is a difference in baseline results between medical students and general population.

More comments:

Positive comments:

It is an original and relevant topic, I think it could be useful to help people observing their emotional state. And an interesting result on baseline score of DAST scale in medical students compared to the general population.

Major comments:

1.What is the main question addressed by the research?

The aim was to determine if HRV and average heart rate could be used as a predictor of acute stress, and, contribute to the development of a physiological algorithm for stress which could be uti in smartwatch technologies.

I am not sure I understand correctly, did the authors hope for estimating HRV which predicts acute stress? Is it possible? I think this study does not test this hypothesis. It investigates the change of HRV under stress in two groups exposed to acute stresor.

What the authors mean by „acute stress” in this study. It should be explained.

Is resting HRV considered as HRV just before the TSST? Meaning is it considered as baseline HRV or in anticipation of stress.

Do you mean that absolute value of HRV could predict that someone will react to stress? I do not understand.

If HRV is to predict stress what would be the time where it changes from baseline. Is it a part of stress reaction?

The authors also cite papers investigating HRV under stress and they conclude the results are inconclusive although they say: "Studies generally support an increase in HF HRV during acute mental stressors, owing to an increase in parasympathetic-oriented 247 concentration during stress tasks.”

I believe such statement needs citation for instance.

Kim HG, Cheon EJ, Bai DS, Lee YH, Koo BH. Stress and Heart Rate Variability: A Meta-Analysis and Review of the Literature. Psychiatry Investig. 2018 Mar;15(3):235-245. doi: 10.30773/pi.2017.08.17. Epub 2018 Feb 28. PMID: 29486547; PMCID: PMC5900369.

2.The aim of the study should be modified. As I understand it investigates the difference in response to acute stresor in two groups.

I also suggest elaborating more on the baseline difference in DASS score and linking it with chronic stress concept. Is it possible that there was a bias because students were tested in a familiar environment? Also where there any other reasons for general population group to have higher DAST score? This should be described in limitations paragraph.

I cannot find information on what was the time frame of ECG recording.

3.In discussion first the finding directly connected to the aim should be desired. Conclusions are not clear and do not correspond with the aim of the study. It is not clear what the authors were looking for.

4. references :  The limitation section is missing - the group is small.

Author Response

Thank you for your very helpful critiques. 

Round 2

Reviewer 1 Report

Reviewer's Comment to V2. Several changes have been made in the manuscript which have improved it, but there are still a few things that could be improved.  It still seems that the significant contribution of this paper is to the general field of algorithm development for stress evaluation, which could inform the use of any type of wearable technology.  The emphasis on FitBit in the title may cause some people to overlook the paper. 

General comments:

“Clinical cohort” versus “medical students”: The manuscript still contains reference to a “clinical cohort” which indicates a patient population when they in fact mean the medical students. The manuscript should be searched for the use of “clinical cohort” and the term should be replaced.

There are numerous typographical errors (extra spaces, missing punctuation).

Methods:

Use of Fitbit

Author’s response: “I agree that the HR data could have been collected from the ECG, though we decided to use Fitbit as it was the wearable device of interest to the researchers and has been reported as accurate within the literature. Analysis was not made between the ECG vs FitBit, though this could’ve been attempted; My understanding was that Fitbit is validated for recording HRV quite accurately? “

Reviewer’s response: It is the author’s responsibility to provide a reference or data documenting that the FitBit Versa 2 is validated for recording HRV. Recent reviews of which I am aware either do not include it or do not provide detailed information about the validity of this model. The authors should explain why the FitBit was of interest. See:

Fuller, D., Colwell, E., Low, J., Orychock, K., Tobin, M. A., Simango, B., Buote, R., Van Heerden, D., Luan, H., Cullen, K., Slade, L., & Taylor, N. G. A. (2020). Reliability and Validity of Commercially Available Wearable Devices for Measuring Steps, Energy Expenditure, and Heart Rate: Systematic Review [Review]. JMIR Mhealth Uhealth, 8(9), e18694. https://doi.org/10.2196/18694

Hinde, K., White, G., & Armstrong, N. (2021). Wearable Devices Suitable for Monitoring Twenty Four Hour Heart Rate Variability in Military Populations. Sensors (Basel, Switzerland), 21(4), 1061. https://doi.org/10.3390/s21041061

Nelson, B. W., Low, C. A., Jacobson, N., Areán, P., Torous, J., & Allen, N. B. (2020). Guidelines for wrist-worn consumer wearable assessment of heart rate in biobehavioral research. npj Digital Medicine, 3(1), 90. https://doi.org/10.1038/s41746-020-0297-4

The comments on the FitBit community blog seem to express concern about HRV validity (see: https://community.fitbit.com/t5/Other-Versa-Smartwatches/How-accurate-is-HRV/td-p/4499184)

Lines 152-155.  Was data from the Fitbit used to assess HRV or heart rate? The original manuscript implied the Fitbit data was used to collect heart rate data. Line 153 of the revised manuscript states “heart rate variability”. Which data were collected using the Fitbit? 

Results:

Tables 3.3 and 3.4, It might help to clarify things if the table footnotes indicated which device ECG or Fitbit was used to collect the data presented in the table.

Discussion:

Reviewer’s comment to version 1: The discussion should begin by focusing on the main purpose of the article, which was to inform the development of an algorithm. The data comparing the medical student and general population cohorts are of interest, but are not the primary purpose of the article.

Author’s response: Discussion began by noting significant findings of Dass anxiety correlation with LF and overall findings of LF increase in acute stress

Reviewer’s comment to version 2:  

Providing a brief summary of the main results at the beginning of the discussion is good, but the first paragraph should also summarize why the study was done. What gap were you trying to fill? Why is this paper important? See the following webpage https://www.biosciencewriters.com/How-to-Write-a-Strong-Discussion-in-Scientific-Manuscripts.aspx

Author Response

Attached with many thanks.

Thank you for your careful review of our paper. Your expertise were greatly appreciated.

Reviewer 2 Report

The aim in the abstract and introduction is not modified (verses 29 and 84). 

Author Response

As per attachment.
Thank you very much for your reviews of our paper.
